# Fair curve designing by Said-Ball curve

**Sana Zafar, Maria Hussain** *

Department of Mathematics, Lahore College for Women University, Lahore, Pakistan

* maria.hussain@lcwu.edu.pk

## Abstract

Fair curves are visually alluring curves and are free of unnecessary design features. A new curve designing method is introduced using the tangential continuous rational cubic Said-Ball curve. Fair curves are achieved by controlling its length and variation in curvature. It has enough degrees of freedom (control points and free parameters). A family of curves can be obtained for different choices for the values of free parameters. The control points are fixed by employing the $G^1$ continuity conditions at the end points of the RCSBC. The optimal values of the remaining free parameters that are the weights of the underlying curve are obtained by constructing the optimization problems with stretch energy and curvature variation energy as the objective functionals. Different numerical examples are constructed to consolidate the effectiveness of the presented methods. Finally, two applications of the proposed techniques is also presented in the end.

## 1. Introduction

Fair curves have earned significant admiration in the past several years, specifically due to their enormous role in industrial design. Their applications can be seen in the automobile industry in the roof designing of cars, aerospace industry for the designing of wings and tails of aircraft, highways and railway tracks, self-driven cars, and domestic machinery. This wide range of applications drew the attention of researchers to the study of fair curve generation. The curves with minimum energy are referred to as fair curves. The importance of constructing fair curves by minimizing their energy over the prior methods can be easily seen. Algorithms of the proposed methods are simple, and their computation time is considerably small. Comparison of the proposed fair curve designing methods to the existing techniques is given in Tables 3 and 6.

Various techniques have been implemented for achieving fair curves. Ahn et al. [1] proposed a technique of controlling the length and bending energy of the quadratic Bézier curve. In [2], the authors proposed an energy model for algebraic splines, i.e., A-spline, considering both stretch and strain energy. Seeking an energy minimizing curve yielded in complex integrals which were solved only by numerical methods.

**Data availability statement:** All relevant data are within the manuscript.

**Competing interests:** The authors have declared that no competing interests exist.

For easy handling and computation, the authors introduced a simpler expression of energy which was faster to compute and yet gave accurate results. Brunnett and Kiefer [3] discussed a method to compute minimum energy spline with the idea that the end point tangent vectors were known. Cantón, Fernández-Jambrina and Vázquez-Gallo [4] derived the conditions for the monotonicity of the curvature of the planar Bézier curves. Choi, Curry and Elkaim [5] proposed a method of minimizing the maximum magnitude of curvature by finding the optimal control lengths of the quadratic Bézier curve. Gang, GuoZhao and WenYu [6] discussed three types of energies of the Bézier curves. The energy functionals were represented in terms of control points. The idea was based on as some of the control points are given and the remaining control points are evaluated by minimizing the energies of the Bézier curve. Habib and Sakai [7,8], presented different transition curves for shape control. Jaklič and Žagar [9,10] proposed the construction of optimal planar cubic Hermite curve by minimization of curvature variation energy and strain energy. In both papers, the authors used $G^1$ interpolating conditions. Johnson and Johnson [11] presented a method of producing fair curve that interpolates a set of control points. The tangent directions at the interpolating points were kept variable and the idea was to obtain them by minimizing the energy of the curve. Juhász and Róth [12] proposed a method of moving control points to obtain minimum energy curves. The shape of the curve was obtained by controlling the strain energy functional while moving of some of the control points and keeping rest of them as fixed. This concept of movable control points to achieve the fair curve was a time-consuming process which decreased the effectiveness of these methods. Li and Zhang [13] presented a $G^1$ fair cubic Hermite curve by studying the stretch and jerk energy of the cubic Hermite curve. The fair curve was obtained by solving the bi-objective optimization problem. Lu discussed a method of minimizing the jerk energy of a cubic interpolant in [14] and presented a method of minimum strain energy of the planar quintic Bézier curve in [15]. In [16], the authors constructed fair $G^1$-cubic and $G^2$-quintic Hermite curve using the curvature variation energy of the corresponding curves. In [17], the author proposed a method for achieving fair curve. The curve under consideration was expressed in the form of sequence of vertices instead of the basis functions. These vertices were specified by the derivatives upto the third order. The disadvantage of such methods was the curve produced using the discrete set of vertices were not compatible with the CAD software.

Said Mad Zain and Misro [18] used fractional Bézier curve to construct fair and smooth curve using different types of continuity conditions. Saito and Yoshida [19] derived functions of the lowest possible degree to evaluate the curvature monotonicity of planar and spatial rational Bézier curves. These functions were evaluated in the form of Bernstein basis. The curvature monotonicity was evaluated by subdivision and Bézier clipping. Wang, He, Song and Zhao [20] derived sufficient conditions on Class A matrices to produce 2D Bézier curves with monotone curvature profile. Wolberg and Alfy [21] proposed the interpolation of a monotone data set by making use of cubic splines. The aim was to construct a smooth curve that passes through the control points at the same time satisfying the monotonicity conditions. The authors

developed an energy minimization framework for the construction of such smooth curves. Yang [22] introduced Euler Bézier spirals or Euler B-spline spirals which approximated the curvature linearly. Due to specifically defined control points the proposed method interpolated the $G^1$ boundary data by Euler Bézier or B-spline spirals.

The research work proposed in this paper is devoted to the study of constructing fair curves. The rational cubic Said-Ball curve is used for the fair curve construction techniques presented in this paper. The required fair curve is obtained by minimizing the suitable energy functionals. These energy functionals are the stretch energy which is the measure of the curves' length and the jerk energy which is used to control the variation in curvature of the curve to be fair. The optimal values of free parameters are obtained by minimizing these functionals. The energies are compared with the prior techniques [16,23] manifesting the efficiency of the method. Furthermore, practical applications of the presented techniques are also given showing the effectiveness of our method. Example 1–4(stretch energy) are compared to [23] while Example 5–8(curvature variation energy) are compared to [16].

The paper is arranged as follows: Section 2 discusses the tangent continuous rational cubic Said-Ball curve. Section 3 establishes fair curve construction techniques. Section 4 presents numerical examples along with the comparison with prior techniques. Section 5 concludes the paper.

## 2. Tangent continuous RCSBC

Ball introduced the cubic polynomials which reformed the manual lofting techniques of the CONSURF surface lofting programme [24]. It was used to design an aircraft design system [24]. The generalized Ball basis is normalized totally positive, and it possesses the shape-preserving property like Bernstein basis [25]. The recursive algorithm of Said-Ball curve for evaluating a polynomial curve runs twice as fast as the de Casteljau algorithm of Bézier curve, therefore the Said-Ball curve computationally efficient than the Bézier curve [26].

The cubic Said-Ball curve uniquely approximates a function but due to the absence of free parameters, Said-Ball curve does not have the ability to modify the shape of the curve. Therefore, a rational function with two free parameters is introduced for the construction of fair curve. It is a special case of rational cubic Said-Ball curve [26] used to represent conics. The introduced rational cubic Said-Ball curve(RCSBC) $P(t)$, is defined as follows:

$$P(t) = \sum_{i=0}^{3} R_i(t) b_i, \ 0 \leq t \leq 1. \tag{1}$$

$R_0(t) = \frac{\hat{A}_0}{D(t)}$, $R_1(t) = \frac{\hat{A}_1 w_1}{D(t)}$, $R_2(t) = \frac{\hat{A}_2 w_2}{D(t)}$, $R_3(t) = \frac{\hat{A}_3}{D(t)}$, $D(t) = \hat{A}_0 + \hat{A}_1 w_1 + \hat{A}_2 w_2 + \hat{A}_3$. $R_i(t)'s$ are the rational basis functions of the curve (1) and $\hat{A}_i's$ are cubic Said-Ball basis functions, $\hat{A}_0 = (1-t)^2$, $\hat{A}_1 = 2t(1-t)^2$, $\hat{A}_2 = 2t^2(1-t)$, $\hat{A}_3 = t^2$. The control points $b_i's$ and the unknown parameters $w_i's$ are used for shape modification.

By employing tangency constraints, the following control points were determined:

$$b_0 = (0,0), \ b_3 = (x, y), \tag{2}$$

$$b_1 = d_0 \left( \cos \theta_0, \sin \theta_0 \right), \ b_2 = (x, y) - d_1 \left( \cos \theta_1, \sin \theta_1 \right). \tag{3}$$

Substituting the values of control points from (2) and (3) in (1), a $G^1$-continuous rational cubic Said-Ball curve is obtained. Here $d_i's$ are the positive free parameters which are the norms of distances between control points $b_0, b_1$ and $b_2, b_3$. Along with these parameters and weight parameters, we now have four free parameters to be determined.

## 3. Fair curve construction algorithm

Construction of fair curves has been a major problem in curve design environment. Fair curves are showing their importance in various fields such as: aerospace industry, in the manufacturing of automobiles, ship-hulls, autonomous vehicles,

highway designs and domestic appliances. Minimizing the energy of a curve is an interesting method for achieving a fair curve. These energies play a significant role in the final design of the curve. By minimizing these energies, a fair curve can be achieved.

### 3.1 Stretch energy

In curve design environment to achieve fair curve, it is needed to avoid unnecessary oscillations of the curve. These oscillations can be avoided by controlling the length of the curve. This can be done by minimizing the length of the curve which is termed as stretch energy. Mathematically, it can be written as, $E_s = \int_0^1 \|P'(t)\| \, dt$ for a planar curve, $P(t)$ for $t \in [0, 1]$. The integral is difficult to compute as it resulted in complex non-linear terms. Therefore, to avoid complexity its approximate form [6] is used

$$\hat{E}_s = \int_0^1 \|P'(t)\|^2 \, dt. \tag{4}$$

The first derivative $P'(t)$ of the curve, $P(t)$ is given as follows:

$$P'(t) = \frac{\sum_{k=0}^5 (1-t)^{5-k} t^k \widetilde{A}_k}{(D(t))^2}. \tag{5}$$

The expression for $\widetilde{A}'_k s$ can be obtained by simple computation. Substituting the value of $P'(t)$ from (5) in (4). It is seen that the resulting definite integral can only be evaluated by numerical integration. We are using Weddles' rule of numerical integration [27]. After integration, (4) becomes

$$\hat{E}_{stretch} = \sum_{i=1}^{11} \widetilde{S} \, T \hat{S}. \tag{6}$$

Here $\widetilde{S} = \begin{bmatrix} \widetilde{S}_1 & \widetilde{S}_2 \cdots \widetilde{S}_{10} & \widetilde{S}_{11} \end{bmatrix}$,

$T = \begin{bmatrix} S_{1,1} & \cdots & S_{1,6} \\ \vdots & \ddots & \vdots \\ S_{11,1} & \cdots & S_{11,6} \end{bmatrix}$, $\hat{S} = \begin{bmatrix} \hat{S}_1 & \hat{S}_2 & \hat{S}_3 & \hat{S}_4 & \hat{S}_5 & \hat{S}_6 \end{bmatrix}^T$, $\hat{S}_1 = \|b_1\|^2$, $\hat{S}_2 = \|b_2\|^2$, $\hat{S}_3 = \|b_3\|^2$,

$\hat{S}_4 = b_1 \cdot b_2$, $\hat{S}_5 = b_1 \cdot b_3$, $\hat{S}_6 = b_2 \cdot b_3$.

The mathematical expressions for $\widetilde{S}_i$ and $S_{i,j}$ are given in Appendix.

The deduced terms of $\widetilde{S}$, $T$ and $\hat{S}$ clearly depict that they depend only on $d_0, d_1, w_1$ and $w_2$. Therefore, the functional can be rewritten as

$$\hat{E}_{stretch} = \hat{E}(d_0, d_1, w_1, w_2). \tag{7}$$

The desired fair curve is achieved by minimizing this stretch energy for the unknown parameters $d_0, d_1, w_1$ and $w_2$. To ease the complexity of the problem the values of $d_0$ and $d_1$ are evaluated by setting $\frac{\partial \hat{E}}{\partial d_0} = 0 = \frac{\partial \hat{E}}{\partial d_1}$, i.e., when the minimum is reached.

$$d_0 = \frac{-F_1 + A_3 d_1 \cos(\theta_0 - \theta_1)}{2A_1}, \; d_1 = \frac{-(2A_1 F_2 + F_1 A_3 \cos(\theta_0 - \theta_1))}{4A_1 A_2 - A_3^2 (\cos(\theta_0 - \theta_1))^2}, F_1 = (x \cos \theta_0 + y \sin \theta_0)(A_3 + A_4),$$

$$F_2 = -(x \cos \theta_1 + y \sin \theta_1)(2A_2 + A_5).$$

Here $d_0$ and $d_1$ are the norms of distances of control points $b_0$, $b_1$ and $b_2$, $b_3$, i.e., $d_0 = \|b_1 - b_0\|$, $d_1 = \|b_3 - b_2\|$. The $\theta_0$ and $\theta_1$ are the horizontal angles made be the curves at the initial and terminal points. Also, $A_j = \sum_{i=0}^{14} \widetilde{S}_i S_{ij} \hat{S}_j$, for $j = 1, 2, 3, 4, 5, 6$. Now two unknowns are left and the expression can be rewritten as

$$\hat{E}_{stretch} = \hat{E}(w_1, w_2),$$

which we evaluate by the optimization problem-I:

### 3.1.1 Optimization problem-I.

$$Minimize\ \hat{E}_{stretch}(w_1, w_2), \qquad Subject\ to\ w_1 > 0, w_2 > 0 \tag{8}$$

It follows that the objective function in (8) is a non-linear function. For solving the optimization problem-I, we are using MATLAB software. In which the builtin function "fminunc" is used for minimizing this objective function.

The MATLAB built-in function "fminunc" is used to find the minimum of an unconstrained optimization problem. This function is based on the Quasi-Newton algorithm [28] which is a second order method. The optimization technique is better than the gradient descent methods as the method uses the gradient of the function to update an approximation to the inverse Hessian matrix. These methods are efficient in solving problems with non-convex objective function.

The MATLAB "fminunc" finds a minimum of a function with severable variables. The process starts with an initial guess at which the local minimum of the function is computed.

1. Start with an initial guess $x_0$ at the minimum, set $k = 0$. Initialize $H_0 = I$ the $n \times n$ identity matrix.

2. Calculate the gradient of the function $\nabla f(x_k)$ and generate the search direction as $h_k = -H_k \nabla f(x_k)$.

3. Generate the search direction in the direction of $h_k$ and put $x_{k+1} = x_k + \hat{t} h_k$ where $\hat{t}$ minimizes $f(x_k + t h_k)$.

4. Compute $H_{k+1}$ by modifying $H_k$. Set $k = k + 1$ and go to step 2.

5. Repeat steps 2 and 3 until the function converges to a minimum.

## 3.2 Curvature variation energy

In various design processes, gradual change in curvature along a curve is required which is often considered as a curve's fairness measure. Generally, a high degree of fairness is required. This concept of fairness is usually associated with the curvature of the underlying curve. A curve is considered fair if it consists of smoothly varying curvature having a few inflection points and curvature extrema. The curve that satisfies all these requirements is named as minimum variation curve MVC [29].

Mathematically for a planar curve, $p(t)$, $t \in [0, 1]$, with curvature denoted by $\kappa(t)$, this functional can be written as a change in curvature norm [6], $E_{cve}$ as follows:

$$E_{cve} = \int_0^1 \|\kappa'(t)\|^2\ dt, \tag{9}$$

and is known as curvature variation energy. Formula of curvature of parametric curve $k(t) = \left(\frac{dx}{dt}\frac{d^2y}{dt^2} - \frac{dy}{dt}\frac{d^2x}{dt^2}\right)\left(\left(\frac{dx}{dt}\right)^2 + \left(\frac{dy}{dt}\right)^2\right)^{-3/2}$ involves radical terms, which has the computational complexity for evaluation of integral in (9). It also increases approximation error while evaluating the integral involved in (9) by numerical integration technique. Therefore, in literature the approximate form of the curvature variation energy is used which is known as jerk energy [6]. The jerk energy, $\hat{E}_{jerk}$, of the curve $P(t)$ is calculated as follows:

$$\hat{E}_{jerk} = \int_0^1 \left\| P'''(t) \right\|^2 \, dt, \tag{10}$$

$P'''(t)$ is a ninth-degree rational polynomial given in (11).

$$P'''(t) = \frac{\sum_{k=0}^9 (1-t)^{9-k} t^k C_k}{(D(t))^4}. \tag{11}$$

The expression for $C_k$'s can be obtained by simple computation. The following expression is thus obtained after applying Weddle's rule of integration to the expression (10)

$$\hat{E}_{jerk} = \sum_{i=1}^{19} M_i L_{i,j} \hat{L}. \tag{12}$$

$$\hat{L} = \begin{bmatrix} \hat{L}_1 & \hat{L}_2 & \hat{L}_3 & \hat{L}_4 & \hat{L}_5 & \hat{L}_6 \end{bmatrix}^T, \hat{L}_1 = \|b_1\|^2, \ \hat{L}_2 = \|b_2\|^2, \ \hat{L}_3 = \|b_3\|^2, \hat{L}_4 = b_1 \cdot b_2, \hat{L}_5 = b_1 \cdot b_3, \hat{L}_6 = b_2 \cdot b_3,$$

$$L_{i,j} = \begin{bmatrix} L_{1,1} & \cdots & L_{1,6} \\ \vdots & \ddots & \vdots \\ L_{19,1} & \cdots & L_{19,6} \end{bmatrix}$$

Here the complete expressions for the $M_i's$ and $L_{i,j}'s$ are given in the Appendix.

By the values of $M_i, L_{i,j}$ and $\hat{L}$, the expression (12) clearly depends on the positive real free parameters $d_0, d_1, w_1$ and $w_2$.

$$\hat{E}_{jerk} = \hat{E}(d_0, d_1, w_1, w_2). \tag{13}$$

The computation of four unknowns can increase the complexity of the problem. To overcome this complexity, we evaluate the unknowns $d_0, d_1$ by putting $\frac{\partial \hat{E}}{\partial d_0} = 0 = \frac{\partial \hat{E}}{\partial d_1}$. We get

$$d_0 = \frac{-F_1 + A_3 d_1 \cos(\theta_0 - \theta_1)}{2A_1}, d_1 = \frac{-(2A_1 F_2 + F_1 A_3 \cos(\theta_0 - \theta_1))}{4A_1 A_2 - A_3^2 (\cos(\theta_0 - \theta_1))^2},$$

$$F_1 = (x \cos\theta_0 + y \sin\theta_0)(A_3 + A_4), \ F_2 = -(x \cos\theta_1 + y \sin\theta_1)(2A_2 + A_5),$$

where $A_j = \sum_{i=0}^{14} M_i L_{ij} \hat{L}_j$, for $j = 1, 2, 3, 4, 5, 6$. Now we are left with two unknowns which are to be evaluated by the optimization problem written below:

### 3.2.1 Optimization problem-II.

$$\text{Minimize } \hat{E}_{jerk}(w_1, w_2)$$

$$\text{Subject to } w_1 > 0, w_2 > 0$$

The process of minimization of energy functionals, i.e., stretch energy and curvature variation energy can be better explained by Algorithm 1.

 

```
Algorithm 1
Step 1. Input the values of initial and final control points, angles θ₀ and θ₁.
Step 2. Find the values of d₀ and d₁.
Step 3. Evaluate the approximated stretch energy and jerk energy of RCSBC.
Step 4. Attain the optimal values of free parameters w₁ and w₂ in optimization problem-I and optimi-
zation problem-II.
Step 4. Substitute the obtained values of d₀, d₁, w₁ and w₂ in (2), (3) to compute the control points
b₀, b₁, b₂ and b₃.
Step 5. Put the extracted values in (1) to achieve optimal fair rational cubic Said-Ball curve.
```

## 3.3 Numerical results

This section discusses some numerical examples and the obtained results effectively demonstrating the efficiency of the proposed methods. Letting the terminal points $P_0$ and $P_1$ be $(0,0)$ and $(1,0)$ respectively with varying end tangent directions $\theta_0$ and $\theta_1$. The different values for these angles are given in Table 1 for stretch energy and in Table 4 for curvature variation energy. The values of unknowns are calculated by solving the optimization problem-I for stretch energy functional is given in Table 2. Similarly, Table 4 consists of values of unknowns obtained by solving the optimization problem-II. The minimum stretch and curvature variation energies are given in Tables 3 and 6, respectively.

The unknowns obtained after optimizing the stretch energy functional are enclosed in Table 2.

The fair curve plots for the above examples are given in Figs 1–4.

Table 3 provides the minimum energies and computation time for the above examples and comparison to the existing techniques.

The input data for curvature variation energy is given in Table 4.

The unknown control points and the values of free parameters obtained after minimizing the curvature variation energy are given in the Table 5.

The fair curve plots for the above examples are given in Figs 5–8.

Table 6 consists of the obtained minimum curvature variation energies.

**Table 1. Input values(stretch energy).**

| Angles | Example 1 | Example 2 | Example 3 | Example 4 |
|---|---|---|---|---|
| $\theta_0$ | $\frac{-\pi}{3}$ | $\frac{-\pi}{3}$ | $\frac{-4\pi}{9}$ | $\frac{-\pi}{2}$ |
| $\theta_1$ | $\frac{\pi}{3}$ | $\frac{-\pi}{4}$ | $\frac{4\pi}{9}$ | $\frac{\pi}{2}$ |

**Table 2. Results obtained by numerical optimization.**

| Examples | $b_1$ | $b_2$ | $w_1$ | $w_2$ | $d_0$ | $d_1$ |
|---|---|---|---|---|---|---|
| Example 1 | $(0.1, -0.1732)$ | $(0.9, -0.1732)$ | $10^3 \times 1.2028$ | $10^3 \times 1.2028$ | 0.2 | 0.2 |
| Example 2 | $(0.1676, -0.2903)$ | $(0.7103, 0.2897)$ | $10^3 \times 1.1077$ | $10^3 \times 1.1074$ | 0.3352 | 0.4097 |
| Example 3 | $(0.0042, -0.0239)$ | $(0.9958, -0.0239)$ | $10^3 \times 1.2099$ | $10^3 \times 1.2099$ | 0.0242 | 0.0242 |
| Example 4 | $10^{-17}(0, -0.8512)$ | $(1, 0)$ | $10^3 \times 1.1344$ | $10^3 \times 1.1344$ | $2.13 \times 10^{-22}$ | $2.133 \times 10^{-22}$ |

**Table 3. Approximated energies and computation time.**

| Examples | Example 1 | Example 2 | Example 3 | Example 4 |
|---|---|---|---|---|
| Approximated stretch energies by the proposed technique | 0.3086 | 0.3587 | 0.5503 | 0.5970 |
| Approximated stretch energies by the technique [23] | 1.1692 | 1.1290 | 1.1999 | 1.2037 |
| Time elapsed in (secs) for the proposed method | 0.037413 | 0.037388 | 0.035363 | 0.033423 |
| Time calculated in (secs) [23] | 0.965043 | 0.166338 | **0.033984** | 0.812080 |

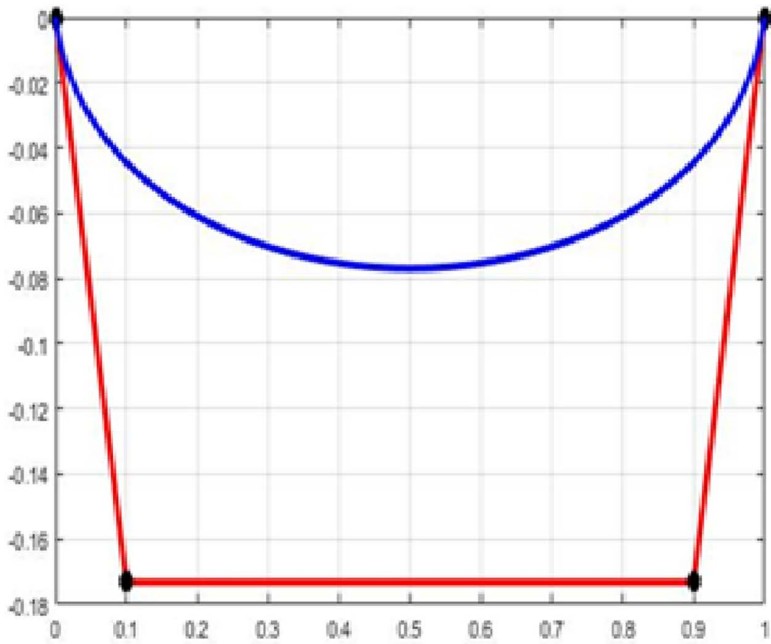

**Fig 1. Approximation curve for Example 1.**

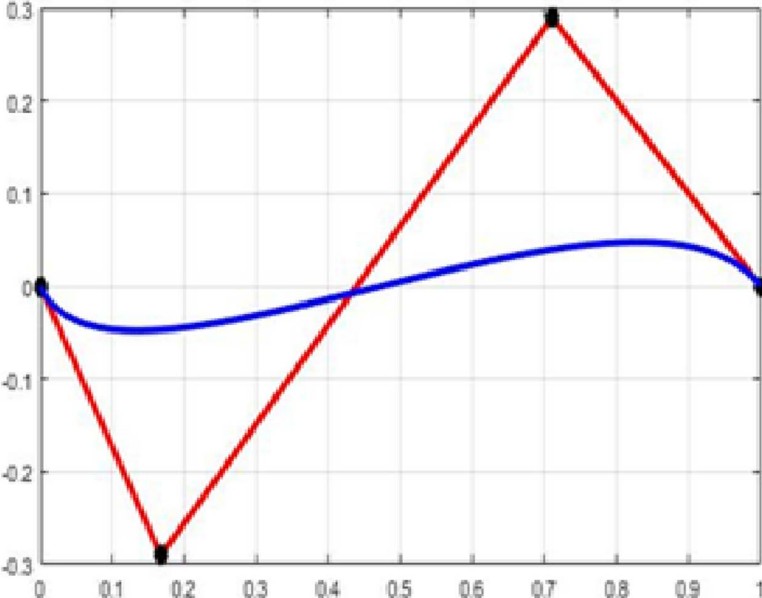

**Fig 2. Approximation curve for Example 2.**

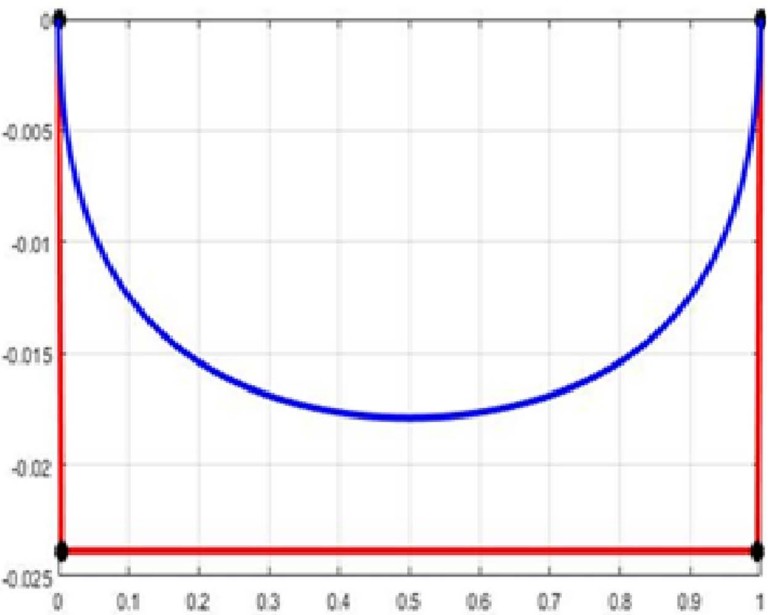

**Fig 3. Approximation curve for Example 3.**

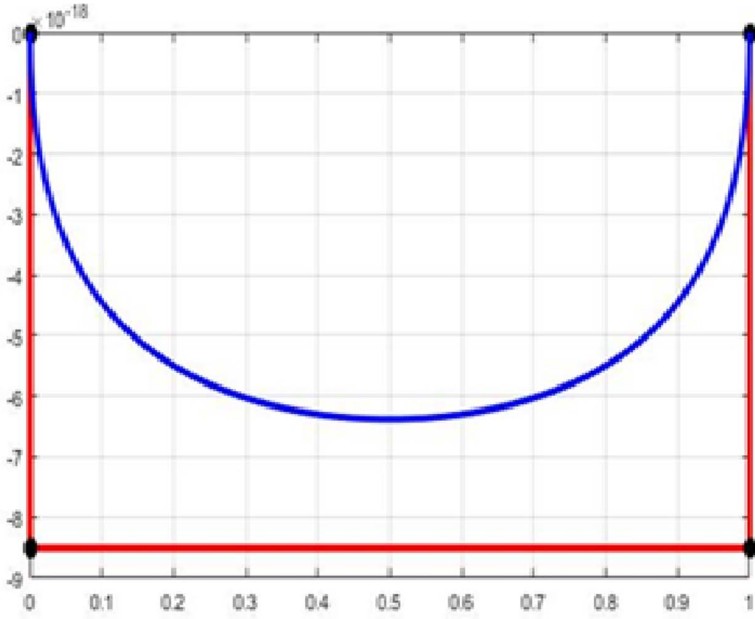

**Fig 4. Approximation curve for Example 4.**

In the end we present an application of the above proposed method for practical use by constructing a glass. The surface of glass is constructed by using the control points and unit tangents at these points. The construction of the glass is hereby showing the efficiency. The initial data for this construction is given in the Table 7. The profile curve of glass consists of four piecewise cubic curves, see Fig 9. The surface of glass is generated by revolving this profile curve about z-axis, see Fig 10.

An example of visually pleasing shape for the character "C" is established using the rational cubic Said-Ball curve. Initial data for the construction of character "C" is given in the following Table 8.

**Table 4. Input data (curvature variation energy).**

| Angles | Example 5 | Example 6 | Example 7 | Example 8 |
|---|---|---|---|---|
| $\theta_0$ | $\frac{-\pi}{4}$ | $\frac{-\pi}{2}$ | $\frac{-2\pi}{3}$ | $\frac{-\pi}{2}$ |
| $\theta_1$ | $\frac{\pi}{4}$ | $\frac{\pi}{4}$ | $\frac{\pi}{2}$ | $\frac{\pi}{2}$ |

**Table 5. Numerical outcomes.**

| Examples | $b_1$ | $b_2$ | $w_1$ | $w_2$ | $d_0$ | $d_1$ |
|---|---|---|---|---|---|---|
| Example 5 | $(0.5, -0.5)$ | $(0.5, -0.5)$ | 0.0582 | 44.8534 | 0.7071 | 0.7071 |
| Example 6 | $(0.4437, -0.7686)$ | $(0.5562, -0.7687)$ | 0.1585 | 24.0504 | 0.8875 | 0.8876 |
| Example 7 | $(0.5361, -0.9285)$ | $(1, -0.4028)$ | 1.2003 | 1.01 | 1.0721 | 0.4028 |
| Example 8 | $10^{-15} \times (0, -0.9683)$ | $(1, 0)$ | 1.5 | 1.7 | $9.6832 \times 10^{-16}$ | $9.6758 \times 10^{-16}$ |

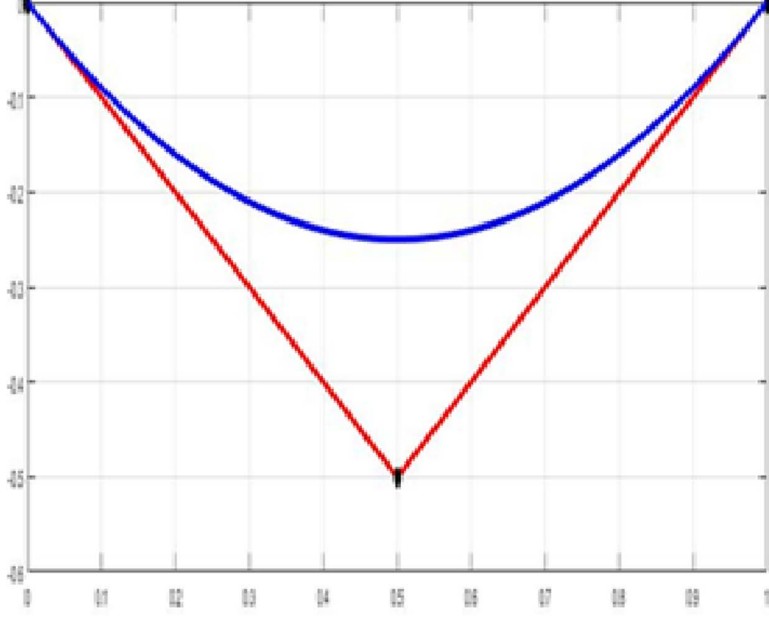

**Fig 5. Approximated curve of Example 5.**

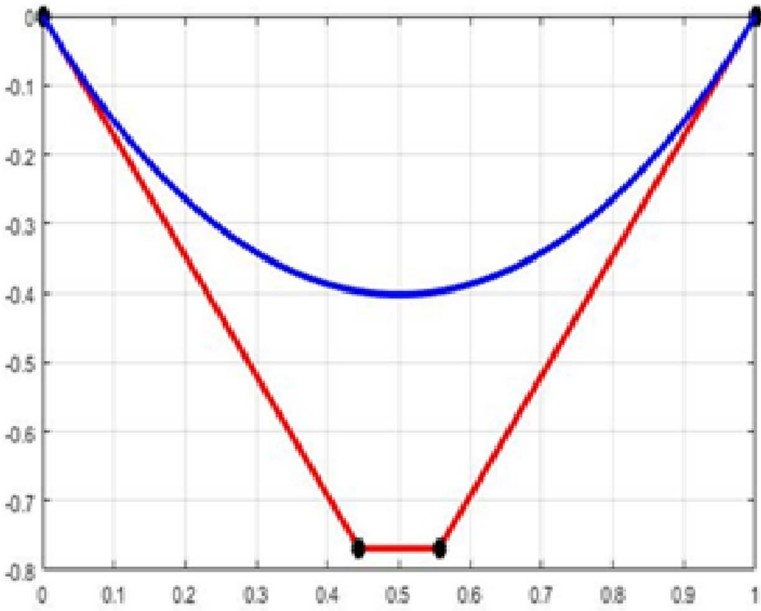

**Fig 6. Approximated curve of Example 6.**

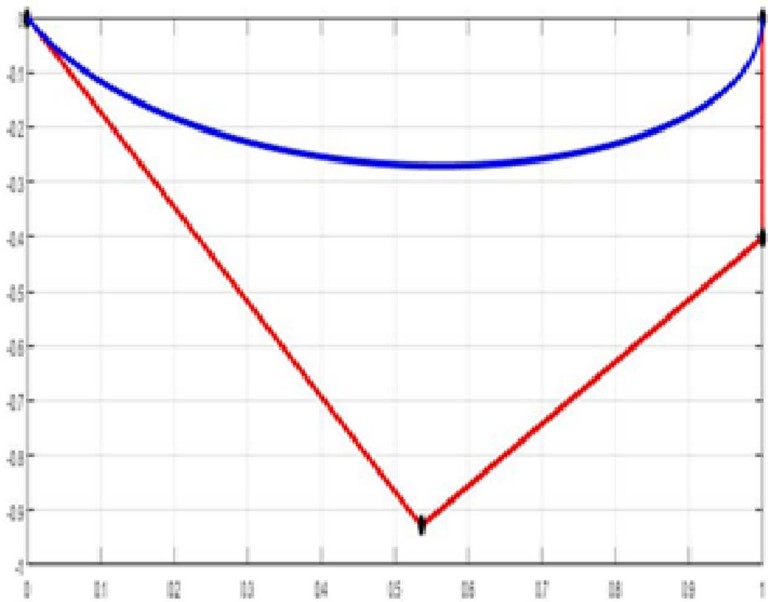

**Fig 7. Approximated curve of Example 7.**

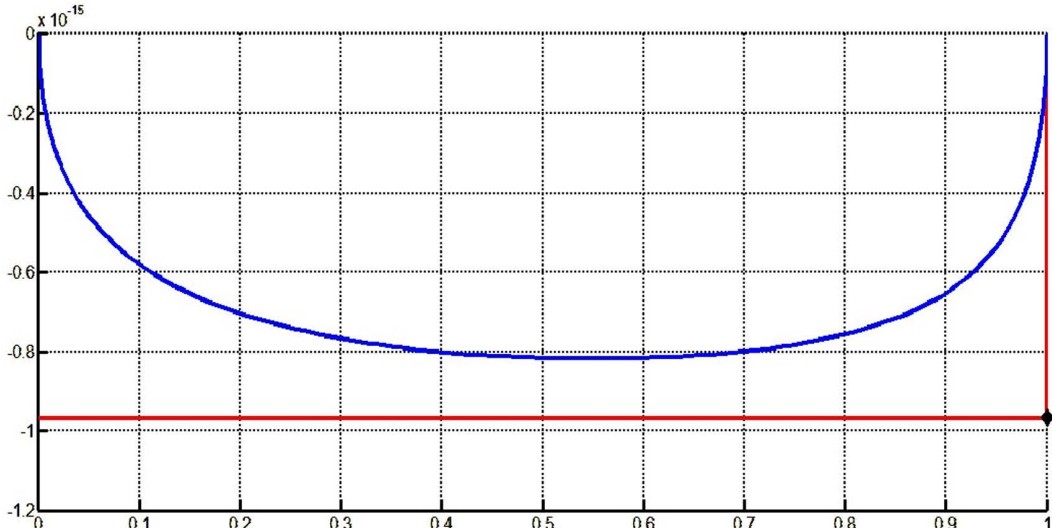

**Fig 8. Approximated curve of Example 8.**

**Table 6. Approximated curvature variation energies and computation time.**

| Examples | Example 5 | Example 6 | Example 7 | Example 8 |
|---|---|---|---|---|
| Approximated curvature variation energies by proposed technique | $1.3248 \times 10^{-14}$ | $5.798 \times 10^{-11}$ | $4.451 \times 10^{-13}$ | 64.9838 |
| Approximated curvature variation energies in [16] | 0.024 | 28.158 | 7.614 | 111.146 |
| Time (secs) elapsed in the proposed method | 0.0528356 | 0.1564283 | 0.0583291 | 0.0491692 |
| Time (secs) elapsed in the method [16] | 0.152 | 0.157 | 0.154 | 0.166 |

**Table 7. Initial data for the construction of glass.**

| Control points | $(1.1, 6)$ | $(1.4, 4.5)$ | $(0.6, 3)$ | $(0.2, 1)$ | $(1.3, 0)$ |
|---|---|---|---|---|---|
| Unit tangents | $(0.743, -0.669)$ | $(-0.122, -0.992)$ | $(-0.766, -0.643)$ | $(0.174, -0.985)$ | $(0.939, -0.342)$ |

The character "C" is made up of two segments, the segment above and below the x-axis. The control points and unit tangents are used to produce the shape of the character "C". The computed values of control points $b_1$ and $b_2$ are given in Table 9. The corresponding final character of font "C" is given in Fig 11.

In Figs 12 and 13, the top and bottom profile of a ruddy deck in flight is approximated respectively by the cubic Hermite interpolation method and the fair curve designing method proposed here. The data of the profile curve of the ruddy duck flight is taken from [27]. It is clear from Fig 12 that cubic Hermite fails to approximate the profile curve of the bird. But the fair curve designing method proposed in optimization problem-I approximated the profile curve approximately.

## 4. Conclusion

A method of achieving fair curve is presented in the proposed research work. $G^1$ continuous rational cubic Said-Ball curve is used for contriving the fair curve. The construction procedure consists of optimization problems of minimizing two objective functions which make the curve fair. Comparison of the metric of the proposed method(stretch energy and curvature variation energy) to the existing methods [16,23] is given in Tables 3 and 6. It shows that the methods proposed in this

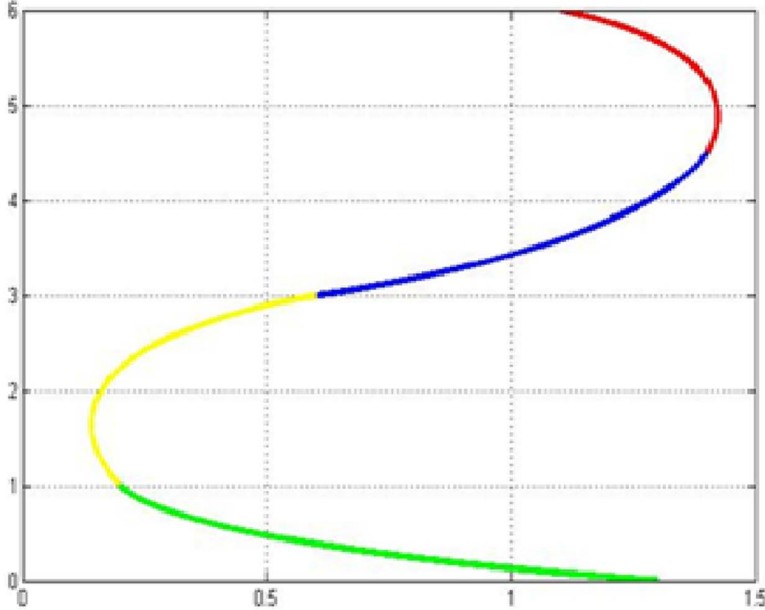

**Fig 9. Profile curve of glass.**

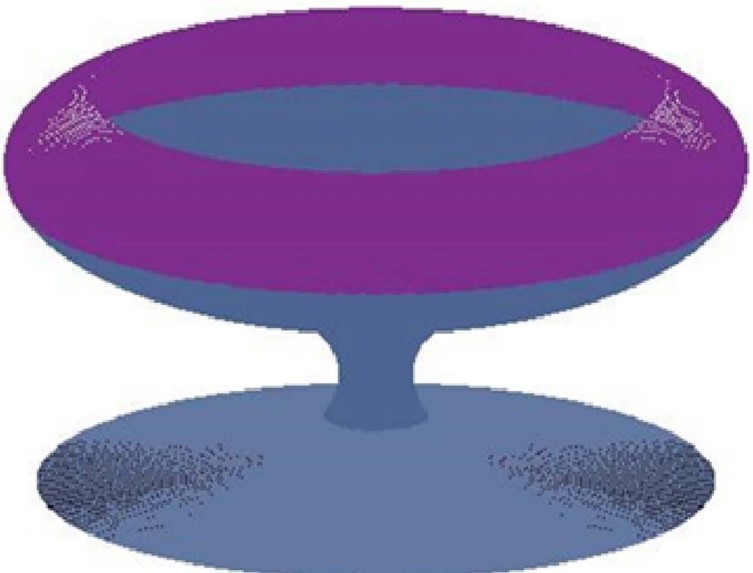

**Fig 10. Surface of revolution of glass.**

**Table 8. Initial data for the construction of character "C.".**

| Control points | $(-0.9999, \pm0.0199)$ | $(0.0050, \pm1.9999)$ |
|---|---|---|

**Table 9. Computed values of control points.**

| $b_1$ | $b_2$ |
|---|---|
| $(-1, \pm 2.7320)$ | $(0, \pm 3.9999)$ |

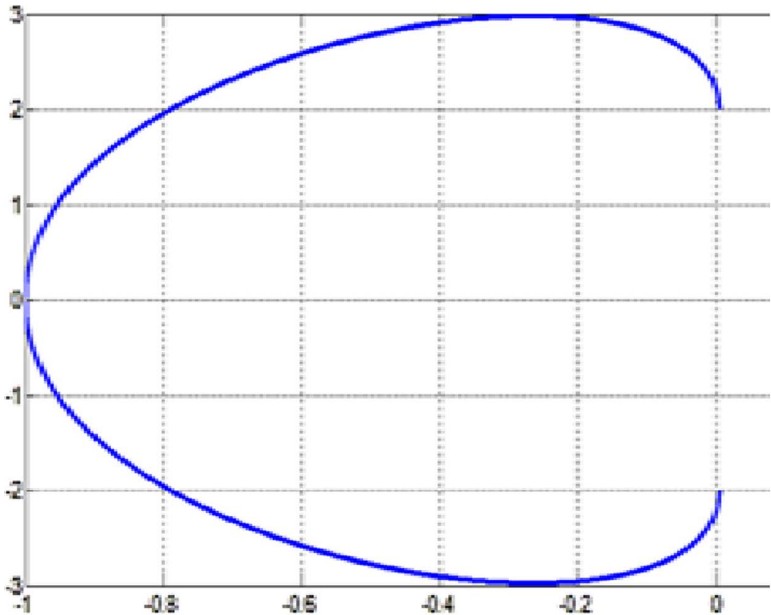

**Fig 11. Character font "C".**

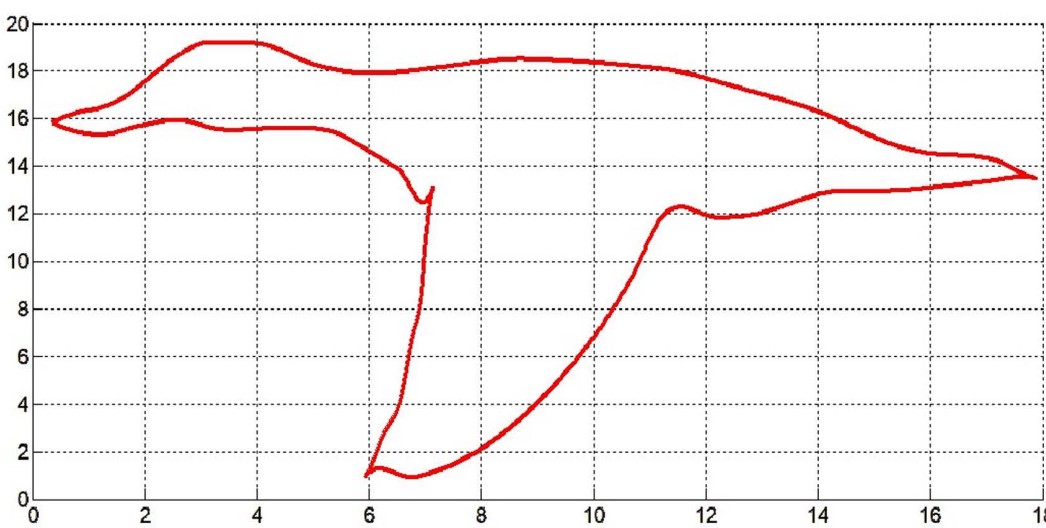

**Fig 12. Cubic Hermite interpolation of profile curve of ruddy deck in flight.**

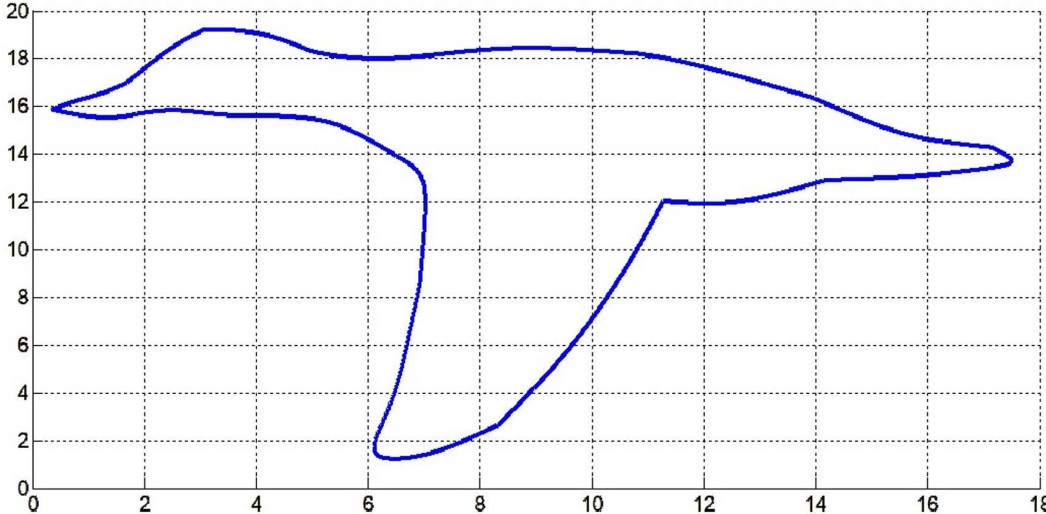

**Fig 13. Fair curve designing of profile curve of ruddy deck in flight by optimization problem-I.**

research paper have less stretch energy and curvature variation energy than the existing methods. Moreover, it works in fractions of seconds, so it is efficient as well.

## Supporting information

**S1 File. Appendix.**
(DOCX)

## Author contributions

**Conceptualization:** Sana Zafar, Maria Hussain.

**Investigation:** Maria Hussain.

**Project administration:** Maria Hussain.

**Supervision:** Maria Hussain.

**Visualization:** Maria Hussain.

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
