## [Decision Letter · Decision Letter 0]

Dear Dr. Hussain,

Thank you for submitting your manuscript to PLOS ONE. After careful consideration, we feel that it has merit but does not fully meet PLOS ONE’s publication criteria as it currently stands. Therefore, we invite you to submit a revised version of the manuscript that addresses the points raised during the review process.

Please see the comments from three reviewers below and in the attachment. In addition to the reviewer comments, please ensure that the study and results are placed into appropriate context in the introduction, discussions and conclusions sections. Please also ensure that the manuscript is copyedited thoroughly. Finally, please share the code you used in this study as per PLOS ONE code sharing policies, and indicate where the code has been shared in the Data availability statement (https://journals.plos.org/plosone/s/materials-software-and-code-sharing#loc-sharing-code).

We look forward to receiving your revised manuscript.

Kind regards,

Hanna Landenmark

Staff Editor

PLOS ONE

Journal Requirements:

"Reimbursement of partial publication fee by the institution"

6. Please ensure that you refer to Figure 5, 6, 7 and 8 in your text as, if accepted, production will need this reference to link the reader to the figure.

Reviewers' comments:

Reviewer's Responses to Questions

**Comments to the Author**

1. Is the manuscript technically sound, and do the data support the conclusions?

Reviewer #1: No

Reviewer #2: Yes

Reviewer #3: Partly

2. Has the statistical analysis been performed appropriately and rigorously?

Reviewer #1: I Don't Know

Reviewer #2: Yes

Reviewer #3: N/A

3. Have the authors made all data underlying the findings in their manuscript fully available?

Reviewer #1: Yes

Reviewer #2: Yes

Reviewer #3: Yes

4. Is the manuscript presented in an intelligible fashion and written in standard English?

Reviewer #1: Yes

Reviewer #2: Yes

Reviewer #3: No

Reviewer #1: the authors can be improved the paper with some examples and other related contents in this topic.

2. the abstract is poor, they can improve it.

3. the introduction can be improved by adding more history.

4. the algorithms can be compared with others.

Reviewer #2: See the attachement.

Reviewer #3: summary: The authors propose to fair rational Said-Ball's curves by means of strain energy and curvature variation energy.

Comments:

1. define fair curve based on references: see Moreton's thesis: https://www2.eecs.berkeley.edu/Pubs/TechRpts/1993/5219.html

2. explain the motivation of choosing rational Said-Ball curve rather then NURBS or Beziers.

3. E_s : this is a well known arc length function, but you stated is as stretch energy However, eqn. (4) is squared which is the stretch energy, add refence. Its approximation is stated in Eqn.(5), show how you derived it.

4. Eqn.(5); is it S^ tilde? In appendix, S^ tilde has u_i terms which are not defined.

5. Eqn.(7) has d_0 and d_1 with theta_0 and theta_1; define them.

6. Explain in brief what is 'fminunc' technique.

7. Give reference to Eqn.(9 - 10).

8. Eqn.(12); is it M^ tilde? In appendix, M^ tilde has u_i terms which are not defined.

9. The experiments shows the details of the resulted rational Said-Ball, however comparison to other types of curves are not carried out thus, we are unable to identify the effectiveness of the proposed method.

10. Example of complex shape which involves piecewise curve is necessary to further concrete the proposed method.

**Do you want your identity to be public for this peer review?** For information about this choice, including consent withdrawal, please see our Privacy Policy

Reviewer #1: No

Reviewer #2: No

Reviewer #3: No

---

## [Decision Letter · Decision Letter 1]

Dear Dr. Hussain,

Thank you for submitting your manuscript to PLOS ONE. After careful consideration, we feel that it has merit but does not fully meet PLOS ONE’s publication criteria as it currently stands. Therefore, we invite you to submit a revised version of the manuscript that addresses the points raised during the review process.

We look forward to receiving your revised manuscript.

Kind regards,

Haipeng Liu

Academic Editor

PLOS ONE

Reviewers' comments:

Reviewer's Responses to Questions

**Comments to the Author**

Reviewer #2: All comments have been addressed

Reviewer #3: All comments have been addressed

2. Is the manuscript technically sound, and do the data support the conclusions?

Reviewer #2: Yes

Reviewer #3: Partly

3. Has the statistical analysis been performed appropriately and rigorously?

Reviewer #2: Yes

Reviewer #3: N/A

4. Have the authors made all data underlying the findings in their manuscript fully available?

Reviewer #2: Yes

Reviewer #3: Yes

5. Is the manuscript presented in an intelligible fashion and written in standard English?

Reviewer #2: Yes

Reviewer #3: Yes

Reviewer #2: The suggested issues were solved. I recommend the manuscript for acceptance. The authors modified the suggested modifications and the manuscript is ready for publication.

Reviewer #3: The authors have corrected most of the parts. However the following need further detail for reproducibility:

1. Fig 9a: state clearly the control points of each segment and the continuity at the joint. The proposed method employs stretch energy and curvature variation energy which involves at least twice differentiable (G2), however the segments are joined at G1, which is differentiable only once. Thus the combined piecewise becomes G1 continuous.

2. Similarly, clearly show the details of the construction of Figure 10 and make sure it is G2 continuous. Compare the metrics proposed in [11, 19] with your approach.

The application examples of piecewise curve and surface construction should be made at least G2 continuous, inline with the G2 fairing technique proposed. Currently the application example does not tally.

**Do you want your identity to be public for this peer review?** For information about this choice, including consent withdrawal, please see our Privacy Policy

Reviewer #2: **Yes: ** Abedallah Rababah

Reviewer #3: No

---

## [Decision Letter · Decision Letter 2]

Dear Dr. Hussain,

Thank you for submitting your manuscript to PLOS ONE. After careful consideration, we feel that it has merit but does not fully meet PLOS ONE’s publication criteria as it currently stands. Therefore, we invite you to submit a revised version of the manuscript that addresses the points raised during the review process.

We look forward to receiving your revised manuscript.

Kind regards,

Haipeng Liu

Academic Editor

PLOS ONE

Additional Editor Comments:

The reviewer 4 raised some concerns on the innovation from a comparative perspective. Please systematically search for relevant literature and add the comparison.

Reviewers' comments:

Reviewer's Responses to Questions

**Comments to the Author**

Reviewer #3: All comments have been addressed

Reviewer #4: (No Response)

Reviewer #5: All comments have been addressed

2. Is the manuscript technically sound, and do the data support the conclusions?

Reviewer #3: Yes

Reviewer #4: Partly

Reviewer #5: Yes

3. Has the statistical analysis been performed appropriately and rigorously?

Reviewer #3: N/A

Reviewer #4: No

Reviewer #5: Yes

4. Have the authors made all data underlying the findings in their manuscript fully available?

Reviewer #3: Yes

Reviewer #4: Yes

Reviewer #5: Yes

5. Is the manuscript presented in an intelligible fashion and written in standard English?

Reviewer #3: Yes

Reviewer #4: Yes

Reviewer #5: Yes

Reviewer #3: The explantion of why the proposed method higher curvature variation energy for example 8 as compared to [19] unclear.Why the method fails to produce better RCSB for a simple shape?

Reviewer #4: 1. The authors fail to demonstrate how their approach offers a significant improvement over existing methods.

2. The paper claims that the Said-Ball curve is more computationally efficient than the Bézier curve, but no evidence or benchmarking is provided. It remains unclear why this curve formulation is preferable over other rational Bézier-based curves.

3. While the paper states that the method is computationally fast, it does not compare computation time across different methods.

4. Why is jerk energy used as a substitute for curvature variation?

5. The energy minimization functionals (stretch energy and curvature variation energy) lead to highly complex expressions.

6. The examples (glass and font character "C") are trivial and do not demonstrate any practical advantages in industrial applications.

7. The literature review is outdated and does not include recent advancements (post-2020) in fair curve design.

This paper fails to meet the necessary standards for publication due to the following major issues as discussed above. I regret to reject this paper.

Reviewer #5: Tthe authors have responded adequately to all previous concerns. In my opinion, no further modifications are needed.

**Do you want your identity to be public for this peer review?** For information about this choice, including consent withdrawal, please see our Privacy Policy

Reviewer #3: No

Reviewer #4: No

Reviewer #5: No

---

## [Editor Report · Decision Letter 3]

Fair Curve Designing by Said-Ball Curve

PONE-D-23-25437R3

Dear Dr. Maria Hussain,

We’re pleased to inform you that your manuscript has been judged scientifically suitable for publication and will be formally accepted for publication once it meets all outstanding technical requirements.

Kind regards,

Haipeng Liu

Academic Editor

PLOS ONE
---

## [Editor Report · Acceptance letter]

PONE-D-23-25437R3

PLOS ONE

Dear Dr. Hussain,

I'm pleased to inform you that your manuscript has been deemed suitable for publication in PLOS ONE. Congratulations! Your manuscript is now being handed over to our production team.

Kind regards,

on behalf of

Dr. Haipeng Liu

Academic Editor

PLOS ONE